# Public Health from the Middle-Out: A New Analytical Perspective

**DOI:** 10.3390/ijerph16244993

**Published:** 2019-12-08

**Authors:** Yannai Kranzler, Yael Parag, Nadav Davidovitch

**Affiliations:** 1Office of Population Health, New Jersey Department of Health, Trenton, NJ 08608, USA; 2School of Sustainability, IDC Herzliya, Herzliya 4610101, Israel; yparag@idc.ac.il; 3School of Public Health, Faculty of Health Sciences, Ben Gurion University of the Negev, Beersheba 8410501, Israel; nadavd@bgu.ac.il

**Keywords:** public health, collaboration, health in all policies, obesity, middle actors, middle-out

## Abstract

Obstacles to collaborative public health frameworks such as Health in All Policies continue to emerge. Partnership-based public health programs present opportunities to study how public servants and practitioners address these barriers in real time. To this end, we utilized “Middle-Out,” a socio-technical analytical approach that highlights the importance of Middle Actors-stakeholders positioned between policymakers and grassroots—to policy diffusion, innovation and collaboration in public health. We conducted participatory observation in administrative settings of Israel’s National Program to Promote Active, Healthy Lifestyle, 30 stakeholder interviews and document analysis. We examined two dimensions of impact from the Middle-Out: *Directions of Influence*—Middle-Up, Middle-Down and Sideways, and *Modes of Influence*—Enabling, Mediating and Aggregating. Through Middle-Out’s lens, our analysis transcends visible benchmarks such as legislation and macro-level resource-allocation, focusing, instead, on elusive administrative spaces within which Middle Actors shape policies, steer funding and facilitate continuity. Incorporating Middle-Out into public health’s conceptual toolbox, we conclude, can improve understanding of complex public health policy arenas, increase recognition of critical socio-technical changemakers and catalyze more effective design of policy tools and strategies that specifically harness Middle Actors’ strengths and qualities.

## 1. Introduction

Governments, health professionals and civil society stretch sectoral comfort zones by addressing determinants of health in diverse policy and practice arenas. Public health’s homegrown governance frameworks, such as Health in All Policies (HiAP) and Healthy Cities, have vastly broadened the scope of the literature’s definition of “health,” and expanded the network of individuals and agencies that practitioners perceive to be potential partners in health [1].

As the challenges of intersectoral public health governance emerge, though, public health scholars increasingly call for the use of innovative approaches for analyzing collaboration over time, including theoretical insights from disciplines characterized by complexities analogous to public health [2,3,4]. The following paper proposes one such approach, the “Middle-Out” socio-technical analytical perspective. Introduced by Janda and Parag [5] and Parag and Janda [6] to support the analysis of energy decarbonization, Middle-Out clarifies and highlights the role of Middle Actors-stakeholders positioned between policymakers and the grassroots—as agents of change. Its lessons, they explain, can be critical in all socio-technical contexts, such as public health.

This paper utilizes the Middle-Out framework’s insights and applies them to analysis in public health contexts. To illustrate the potential of this innovative perspective in public health, we present Israel’s intersectoral National Program to Promote Active, Healthy Lifestyle (henceforth: the NP), adopting two of Middle-Out’s conceptual dimensions of impact: Directions of Influence (Middle-Up, Middle-Down and Sideways) and Modes of Influence (Enabling, Mediating and Aggregating). Our analysis was conducted as part of a larger ethnographic study of Health in All Policies implementation in Israel, including participatory observation in the NP’s administrative contexts, 30 semi-structured interviews with key stakeholders and analysis of legislative documents, policy papers and other materials.

By adopting a Middle-Out lens, our analysis transcends visible benchmarks such as legislation and macro-level resource-allocation, focusing, instead, on the elusive bureaucratic spaces within which Middle Actors shaped policy formulation, steered funding and protected programmatic continuity under conditions of financial and political instability. Integrating Middle-Out into public health’s conceptual toolbox, we argue, can enable a more accurate understanding of complex public health policy arenas, increase recognition of overlooked Middle Actors that serve (or could serve) as agents of socio-technical change and catalyze more effective design of policy tools and strategies that specifically harness Middle Actors’ strengths and qualities.

## 2. From *Why* to *How*

Governments worldwide have embraced intersectoral approaches to public health [7,8]. Obstacles, though, continue to emerge. Health authorities struggle to translate knowledge of the social determinants of health, such as poverty, environmental injustice and housing insecurity into upstream or scalable action [9], often falling back on healthcare interventions and public awareness campaigns [10]. The discourse, having previously shifted from *why* pursue partnership-based public health to *what* steps to take, has shifted once again, this time, to interrogate *how* to promote these steps successfully, with a near-plea to focus on implementation [11,12]. Middle-Out conceptualizes how Middle Actors drive long-term, multi-sector processes. As such, it is a how-centric perspective.

## 3. Middle Actors

The capacity to sign legislation, allocate macro-level resources and use political “bully pulpits” reflect the explicit power of Top-Down. Bottom-Up action derives its influence from the legitimacy, passion and real-world experience at the grassroots level that diffuse upward toward actors positioned higher in the policy hierarchy. Located between the Top and Bottom, though, are networks of organized citizen groups, professional associations, faith-based organizations, and other institutionalized agents of change characterized in the literature as “Middle Actors” [5,6]. To this group we add mid-level public servants that work in government agencies or local authorities who represent, advise and carry out decisions by higher-level policymakers (Top Actors), while also engaging with, seeking advice from and coordinating between individuals and organizations at the grassroots level (Bottom Actors). Middle Actors’ specific contributions tend to be overlooked as they are neither the policy target group nor the policy signatory. As such, they are less visible than those at the Top and Bottom.

It should be noted that what constitutes “Middle Actor” is more than a mere aggregation of bottom actors, or a position between hierarchical levels. Middle Actors have detectable, distinct and defined institutions and organizational structures, including membership, as well as official or unofficial procedures or rules; access to unique material resources such as funding and equipment; established communication channels with their members and other Bottom, Middle and Top Actors; as well as non-material resources such as professional, spiritual or ethical legitimacies. These grant Middle Actors the authority, ability and power to connect, represent, negotiate, and make decisions.

The Middle-Out approach leverages in-betweenness as a space through which Middle Actors combine strong familiarity with and access to the policies, programs, procedures and stakeholders that shape individuals’ lives, as well as the communities they serve [6]. When results depend upon sustaining progress through election cycles and navigating processes which fall between bureaucratic cracks, Middle Actors utilize administrative tools and agenda-setting practices, acting similarly to Kingdon’s “Policy Entrepreneur” [13], Bardach’s “Fixer” [14] and, as Parag and Janda [6] note, “Intermediaries.” In public health literature, Middle Actors often reflect the Organizational/Institutional dimension of the Social-Ecological Model, in between the Interpersonal and Community levels [15].

Parag and Janda introduce diverse examples of Middle Actors, including building professionals, faith-based organizations and commercial property owners [5,6]. Scholars have applied Middle-Out to professional groups such as providers of housing refurbishment [16], heating engineers [17], facilities managers [18], social housing providers [19] and stakeholders facilitating more effective energy storage [20]. Others have focused on Middle-Actor-ship in promoting sustainable energy behavior of community-based organizations [21], municipal officials and non-formal social groups [22,23]. While these works primarily identify Middle Actors’ centrality in environmental and energy policy spheres, Parag and Janda continuously emphasize Middle-Out’s potential in other disciplines characterized by socio-technical determinants, as well.

Notably, while many different types of actors are located between top and bottom actors, not all of them want or can act in a middle-out manner. This could be due to lack of effective channels and resources to effectively connect with and influence Top, Bottom or other Middle Actors, or due to lack of will or interest to connect, act and influence.

## 4. Directions of Influence

Situated between decision-makers and communities, Middle Actors’ influence is multi-directional, impacting policies, strengthening grassroots efforts and facilitating inter-agency collaboration. Parag and Janda [6] refer to these directions of influence as Upstream, Downstream and Sideways. Instead, we use the terms Middle-Up, Middle-Down and Sideways, respectively, to differentiate Middle-Out from public health’s traditional usage of “upstream” and “downstream,” which contrasts between the social determinants of health (upstream) versus bio-medical (downstream) factors and interventions [10]. Middle-Up, therefore, refers to Middle Actors’ influence on policymaking, Middle-Down includes actions that organize and/or elevate grassroots efforts and Sideways represents activities that forge and sustain cooperation with similarly positioned actors across departmental, organizational or sectoral borders [6].

Figure 1, adapted from Parag and Janda [6], illustrates Middle-Actors’ multi-directional influence:

Figure 1 compares between Top-Down, Bottom-Up and Middle-Out actions. If Top-Down embodies authority and Bottom-Up encompasses the potential of numerous passionate individuals to inspire change, Middle-Out capitalizes on Middle Actors’ position in-between. Middle Actors often have direct access to actors and processes at the Top and Bottom of their fields of influence and, importantly, to other Middle Actors who enjoy access to their own respective “Tops” and “Bottoms.”

## 5. Modes of Influence

Within these directions of influence, Janda and Parag [5] propose three modes of influence through which Middle Actors act as agents of change: Enabling, Mediating and Aggregating. *Enabling* includes opening new policy gates and solidifying relationships with those that operate them by providing information, identifying funding sources, drafting legislation and establishing relationships. *Mediating* emphasizes Middle Actors’ negotiating role, translating between professional and sectoral languages and troubleshooting disputes. In public health, they integrate epidemiology, community needs and an understanding of their agencies’ political landscape, set the terms of policy debates by choosing the evidence to present at the Top, maintaining coalitions with their allies in the Middle and communicating with the public at the Bottom. Finally, *Aggregating* recognizes Middle Actors’ potential to affect scalability, including integration of conclusions from a pilot systematically, incorporation of a politician’s agenda or institutionalization of a spontaneous grassroots initiative.

## 6. Israel’s National Program to Promote Healthy Lifestyle

The next sections of this paper utilize Israel’s National Program to Promote Active, Healthy Lifestyle (henceforth: the NP) to illustrate how Middle-Out’s two dimensions of impact, Directions of Influence and Modes of Influence, draw attention to critical processes and actors that drive intersectoral public health practice.

Israel launched the NP in 2011 to address obesity and chronic disease, the country’s leading causes of death. The Ministries of Health (MOH), Education (MOE) and Culture and Sport (MOCS) declared shared leadership. The program’s implementation was led by MOH’s Department of Health Promotion, already responsible for addressing barriers to healthy lifestyle, MOE’s Health Monitor, traditionally responsible for meeting needs such as students’ medication and food allergies, and MOCS’ Department of Women’s Sports and Sports-for-all, a department focused on physical activity at the community level. Additional government ministries, municipalities, the food industry and civil society partnered, as well [4].

The program identified three goals: Improve nutritional intake, increase physical activity and limit obesity. Objectives emphasized closing inequalities. Strategies were informed by World Health Organization (WHO) recommendations, coordinated with Israel’s Healthy Cities Network, and based on the “Healthy Israel 2020” initiative’s recommendations [4].

Through the NP, healthier foods were mandated at all schools, workplaces became incentivized through tax breaks to serve healthier foods, sodium was reduced by at least 30% in over 100 heavily consumed processed foods and new nutritional labeling was piloted on breads. Workshops and consultations were sponsored for individuals with chronic disease and children at risk, with increased funding for low-income residents. NGOs and municipalities launched hundreds of local initiatives, including community gardens and youth bike groups. Educational materials gained exposure through social media [24].

At the structural level, MOCS expanded its focus beyond the athletic elite, increasing investment in youth sports, public exercise facilities and activities for the general population. MOE adopted students’ health as a ministerial goal, tripled its health promotion staff and committed to seeking “health-promoting” status for the entire school system by 2020.

On the other hand, legislative initiatives like limiting junk food marketing to children and calorie-labeling at restaurants were unsuccessful. National and political instability disrupted the program’s consistency, effectively leaving the program without an operable budget from 2013–2016. Budgetary commitments to municipalities were delayed and at times, broken completely. As the budget shrunk, social marketing initiatives ceased. Local programs suffered. The budget was renewed in full in 2017 [25].

## 7. Methods

The analysis presented below was conducted as part of a larger ethnographic study of Health in All Policies implementation in Israel, including participatory observation in the NP’s administrative contexts, 30 semi-structured interviews with key stakeholders and analysis of legislative documents, policy papers and other materials. Due to the program’s complexity, triangulation was critical to enabling multi-dimensional views of the NP’s design and management. The study therefore combined methods (observation, interviews and document analysis) and sought out diverse data sources to facilitate distinct perspectives. For example, interviews included high, mid and entry-level public servants, including politicians, political appointees and professional-level employees in government, civil society and academia.

Observations were conducted in multiple settings, including Parliament, ten government ministries (Health, Education, Culture and Sport, Finance, Economy, Environmental Protection, Welfare and Social Services, Defense, Communication and Interior), NGOs such as the Joint Distribution Society and the National Association of Community Centers, academic institutions and private companies, such as food corporations and private health promotion firms. This included participation in over 500 meetings specifically addressing the NP’s activities. Fieldnotes were taken in real time by hand during formal meetings. Additional sources of observational data included conversations before and after the NP’s meetings, which, in contrast to interviews, allowed for more spontaneous perceptions regarding the issues at hand. Longer, more formal conversations were recorded, with permission. Informal conversations were documented in a research journal up to three days from when they occurred. Fieldnotes were subsequently typed and saved together with interview transcripts and documents, so that they could be incorporated in coding and analysis. The participant-observer defined specific themes in advance to direct the focus of fieldnotes. These included themes related to the implementation of Health in All Policies, such as *collaboration*, *barriers*, *social determinants* and *mechanisms*.

Two sampling strategies guided the interview process: Twenty informants were selected according to a stratified purposeful sampling strategy, based on their ability to contribute different perspectives [26]. A snowball approach then yielded an additional ten informants. Interviews were conducted face-to-face in 2016, in locations convenient for interviewees, while observations were conducted between 2013 and 2016. Each interviewee was given a description of the study prior to being interviewed and provided consent. Interviewees included 21 women and nine men. Interviews were semi-structured and followed a pre-determined interview guide. Interviews were transcribed in 2016 and the beginning of 2017.

Table 1 below, includes information about interviewees:

Alongside fieldnotes and interview transcripts, documents, working papers and PowerPoint presentations, as well as informal communications including emails and text messages, were included in the analysis.

Analyzed documents included: Government Executive Decision 3921/237The National Program to Promote Active, Healthy Lifestyle—MOH Year 1 ReportThe National Program to Promote Active, Healthy Lifestyle—MOH Year 2 ReportThe National Program to Promote Active, Healthy Lifestyle—2012–2016 Evaluation Report, to be published by MOH in 2019The First MOCS National Survey of Physical Activity in Israel, 2013MOCS Call for New Groups to Participate in the National Program for Children’s Sports in 2014–20152014 Legislation for Oversight on Food Quality and Nutrition in SchoolsTel Aviv University 2016 Participatory Social Marketing Summary for the National Program to Promote Active, Healthy Lifestyle

Interview transcripts, texts from research journals (fieldnotes) and programmatic documents were entered into Narralizer, a qualitative research software compatible with both Hebrew and English texts, which aided in data organization for the sake of thematic analysis. Because the research goals built upon specific categories of knowledge that related to intersectoral health initiatives, data was first analyzed according to Directive Thematic Analysis [27]. Based on this approach, existing HiAP literature informed the establishment of the same *a priori* codes utilized as fieldnote themes, including *collaboration*, *barriers*, *social determinants* and *mechanisms*. These were then broken down into subcategories, which emerged from the data, including *informal* and *formal collaboration*, *hierarchical constraints* and *relationships*. Parag and Janda’s Middle Out perspective was identified during peer debriefing as an existing theoretical framework for investigating both the implementation challenges and successes of the NP that emerged from the data and which directly spoke to both the *a priori* themes and emergent subcategories.

The interviewer/participant observer was an employee at the MOH for the entirety of the data collection period. This presented opportunities, as well as challenges. Many of the individuals interviewed, for example, were colleagues. This facilitated access to research sites, as well as a rapport with research subjects, critical to ethnography [28]. One challenge, though, was the natural hesitance to be critical, whether out of a sense of loyalty or fear of ramifications. A second challenge was the potential for hesitance on the part of those who were not direct colleagues, to accurately portray their experiences vis-à-vis the NP to a member of the MOH. To address the first challenge, the research team debriefed in monthly intervals, and conferred with other colleagues at Ben Gurion University regarding data analysis and conclusions [29]. Triangulation and intertextuality [30] were aimed at addressing the second challenge, ensuring that information reflected multiple lenses. Each method—interviews, document and observation—was susceptible to its own limitations. Cumulatively, though, they enabled a layered and more comprehensive analysis. Finally, deidentification of research participants in data presentation and analysis aimed to enable critical analysis and reporting [30].

## 8. Middle Actors in the Nutrition and Physical Activity Policy Arenas

Our analysis concentrates on four distinct types and groups of Middle Actors that operate in the arena of promoting healthy and active lifestyle: mid-level public servants, municipal officials, public schools and academic institutions.

Mid-level officials in government agencies such as the Ministries of Health, Education, Culture and Sport and others have extensive responsibilities around drafting legislation to be signed by Top Actors, forging inter-agency partnerships and convening workgroups, representing their agencies in statutory committees and councils, distributing funding to local agencies and defining the details of broad policy changes. Their dependence on higher-level officials to enact legislation, allocate larger funding amounts and raise the profile of nutrition and physical activity as policy priorities underscores the fact that they are not Top Actors in this space. Ultimately, though, they are representing powerful, organized institutions, not spontaneous community initiatives. They cannot, therefore, be defined as Bottom Actors, either. Their status in-between these hierarchical positions solidifies their status as Middle Actors.

Schools’ potential to impact children’s health is reflected by the WHO’s emphasis on the importance of adhering to systematic “health-promoting school” models. While school systems are critical toward statewide integration of such a module, each individual school has significant power to operate as Middle Actors in this policy arena, bridging between their communities and the national system. Independent schools follow strictly enforced guidelines determined by government officials, and while they have flexibility to allocate limited funding as they see fit, much of their institutional spending is regulated and pre-determined. Top Actors, indeed, define much of the structure in which they operate. And Bottom Actors, including parents, students and local leaders impact school operations, as well. But with governance responsibilities that impact hundreds of children, schools can be critical middle actors in their communities. Each school makes decisions around which public and private programs and funding opportunities to apply for, how to engage parents and students, how to utilize daily recesses, meals and snack times, as well as whether and how to leverage school grounds for physical activity. Each of these roles can have long-term impacts on children’s health.

Like schools, municipalities’ importance has been emphasized extensively by the WHO and others, most prominently, through the Healthy Cities Movement, given their jurisdiction over local policies that impact health, such as zoning laws, park and recreation policy and management of public spaces such as town squares and community gardens. Alongside these responsibilities, municipalities have unique abilities to convene diverse stakeholders and organize their communities. As mentioned previously, municipalities have contexts where they occupy a Top Actor role with regulatory authority. In practice, though, they act as the institutional representative of their city, town or village in meetings with national government agencies, often lobbying and negotiating for resources and communicating the needs of their communities to national leaders. They, too, meaningfully impact the nutrition and physical activity policy arena as Middle Actors.

Finally, academic institutions have played a critical role in furthering the discourse around addressing obesity and chronic disease through promotion of nutrition and physical activity, as well as both developing and advocating for health equity-focused models such as Health in All Policies and Health Impact Assessment. Dependent often on public funding and without direct decision-making power outside academic contexts, they lack Top-Down authority. And while academic officials have participated extensively in popular and social movements for civil, environment and health rights, their institutional affiliation affords them a status that separates them from Bottom actor-ship. In public health contexts, Middle Actor-ship has been manifest by academics’ ability to evaluate and improve public health policymaking frameworks, unapologetically advocating for more progressive public health policies, using research opportunities to amplify the voices of marginalized communities and representing professional positions without compromises that often characterize government agencies.

## 9. Results: Middle-Out Directions and Modes of Influence in the National Program

Analyses of Top-Down or Bottom-Up contributions to the NP would likely focus on visible successes like legislation, high-level budget allocation, executive orders and statutory committees. Middle-Out’s analytical perspective, though, elucidates how the program’s formation and survival amidst political and budgetary instability were facilitated “in the middle,” by Middle Actors who utilized their access to multiple Directions and Modes of Influence.

Table 2 summarizes the analysis and illustrates how Middle Actors utilized Directions of Influence and Modes of Influence to advance the NP. To demonstrate how “in-betweenness” cuts across the sectoral spectrum, we describe actions by the four examples of Middle Actors discussed in the previous section: mid-level public servants, municipal officials, schools and academia.

### 9.1. Middle-Up

#### 9.1.1. Enable

Various teams enabled the program’s multi-sectoral buy-in, as well as its survival amidst political and budgetary crises, by combining knowledge of the system with existing connections. MOH’s Health Promotion Department developed the aforementioned municipalities pilot by integrating lessons learned from the Country’s’ Healthy Cities Network, to which the Department was a long-time partner and funder, into the policy discussions and documents which dictated the terms and expected deliverables of local programs. A non-governmental organization, the Healthy Cities Network provided training and evaluation support to Israeli municipalities that voluntarily committed to the WHO Healthy Cities model and formed a bridge between municipalities and critical mid-level agencies in MOH, MOCS and MOE.

Similarly, the MOH Nutrition Department and the National Food Service established 11 working groups with food companies’ lead technologists to implement salt reduction targets. Food companies employ leading technologists with the expertise and power to alter salt levels in processed foods. Bringing them to the decision-making table, therefore, enabled the program’s success.

In both cases, mid-level departments at MOH (Health Promotion and Nutrition) leveraged their relationships with other Middle Actors to inform and determine the details of the NP’s local funding support and salt reduction programs. Knowing which actors to bring to the policymaking arena was a critical action from the Middle Out.

#### 9.1.2. Mediate

One of the advantages that characterize middle actor-ship is expertise. Middle Actors represent professional disciplines, and accumulate knowledge through their training, experience and close relationships with communities. In the NP, Middle Actors from the leading government agencies, community-based organizations, municipalities and academia negotiated with decision-makers in their institutions to alter long-standing policies to more accurately reflect current health promotion literature.

For example, MOCS historically focused on training athletic standouts for competitive events. Through the advocacy—the mediation—of the MOCS appointee to the NP, though, the Ministry shifted its focus to “Sports-for-all,” funding local initiatives aimed at broader audiences, as well as populations often marginalized from competitive sports in Israel, such as young girls and the elderly. This approach, advocated by the WHO, was achieved through persistent negotiation between professional level staff (Middle Actors) and leadership (Top Actors), and, ultimately, was reflected in the growth of sustainable funds allocated for “Sports for all.”.

#### 9.1.3. Aggregate

Funding was both limited and unpredictable. To facilitate budget aggregation with partners, Middle Actors in the MOH Department of Health Promotion allocated funds to other government agencies, who, once backed by MOH, secured additional funding from their own agencies’ respective treasuries. For example, MOH diverted three million Israeli shekels of its own to MOE in the NP’s first year to support increased health promotion in schools. MOE then committed an additional three million NIS to the program, tripled its health promotion staff and sponsored health promotion trainings for principals and teachers, entitling the year, “The Year of Healthy Lifestyle.” Similarly, the National Police Force received 150,000 NIS through the NP to improve officers’ health. Mid-level managers in the Police presented MOH’s investment to their own management, presenting a detailed strategic plan for matching the MOH’s funding with Police funding. Police leadership (Top Actors) proceeded by granting an additional million per year and requiring the initiative to include rigorous evaluation to gauge its effectiveness. In each of these instances, Middle Actors from MOH coordinated resource-pooling schemes with their counterparts in other agencies (MOE, the Police), aggregating the limited funds in each agency to form a larger whole.

### 9.2. Middle-Down

#### 9.2.1. Enable

The NP’s Executive Action painted broad policy strokes, but the mid-level staff from MOH’s Health Promotion Department, the MOE Health Monitor’s Office and the MOCS Sports-for-All Department, representing the NP’s leading ministries, were charged with details of resource allocation and dividing responsibilities among partners. Thus, beyond impacting the policy language and decision-making from the middle-up, they became the vehicle through which funding was funneled from the middle-down to partners at the municipal and community levels, enabling a targeted, community-driven approach.

Municipal officials described how access to details enabled cross-pollinating initiatives at the programs level, for example, initiatives for improving living conditions for the elderly with Healthy Cities and directing the Recreation Department to consider health opportunities and implications at public events and community centers. Here, too, these actions were not reflected in policy changes, but, rather, were implemented from the middle-down, enabled by coordinating efforts of Middle Actors.

#### 9.2.2. Mediate

Mid-level officials kept abreast of budgetary developments in their own ministries, translating policy and funding windows into health promotion budgets. The National Program’s budget included significant funds for investing in community action. Mid-level public servants, though, responsible for translating broadly defined funding goals, broke down these goals into platforms specifically geared toward marginalized communities, once again mediating the program’s focus to emphasize support for the populations they deemed most in-need of support. Middle Actors with a knack for getting word of an opportunity and an ability to quickly make a proposal succeeded in taking advantage. In this way, officials at the Ministry of Agriculture leveraged their Minister’s desire for a social justice project, to launch a fruit and vegetable distribution scheme. Through their negotiating with their Minister’s staff and demonstrating, through the data and expertise prevalent among Middle Actors that this initiative would be of political value to the Minister, they succeed in securing funds for specific schools in predominantly low-socio-economic neighborhoods characterized by low levels of fruit and vegetable intake.

At the academic level, the NP’s social media team used their “in-between” status to bridge between communities and government agencies. They used their research expertise to conduct interviews and focus groups about the barriers toward active, healthy lifestyle, synthesizing the information from diverse communities often marginalized from government decision-making discourse. The professors and students from the social marketing team created videos, recordings and written reports with insights from community members, effectively acting as mediator between government and citizen.

#### 9.2.3. Aggregate

MOE’s Health Monitor defines school health policy, and in the NP, led efforts to mandate nutrition standards, using MOH funding to triple the MOE staff overseeing the transition to a health-promoting system. This MOE team integrated healthier birthday standards in all nursery schools with materials developed by the social marketing team, and implemented policies including replacing all school’s sugary beverages with water. Their aggregating power to diffuse social marketing innovations or pilot-proven initiatives into programs implemented in thousands of school classrooms was critical.

At the municipal level, city and town government officials utilized their relationships with local businesses and community institutions to form local alliances, which generated a sense of city and town-wide movements. One northern city’s Strategic Planning Department, for example, created a forum of healthy workplaces, as a collaboration with local large employers and the District Health Office. Another large city, in Israel’s southwest, held free physical activity events in its public beaches every Friday, aggregating the municipality’s jurisdiction over public spaces with community centers’ outreach potential to citizens. Healthy living is often a function of social norms. By aggregating their resources to galvanize large groups of citizens and unify them around healthy living, Middle Actors helped create the sense, locally, that nutrition and physical activity were not only possible, but normative.

### 9.3. Sideways

#### 9.3.1. Enable

The NP’s mid-level managers from MOH, MOE, and MOCS forged partnerships with mid-level counterparts in the Ministries of Agriculture, Finance, and Communications, the National Organization of Community Centers, municipalities and academic institutions like Tel Aviv University’s Social Marketing Program, leveraging working relationships with stakeholders that impact determinants of health.

The school system, for example, was a natural partner for a national health promotion program. But the sprawling, fragmented system challenges those seeking an entry point. Middle Actors, though, through their discipline-specific expertise and experience, have the capacity to identify specific individuals with the potential to be impactful partners. In this case, a mid-level public servant at MOH leveraged longstanding partnerships with the Health Monitor to establish the NP’s lead school partner. The NP’s MOCS representative, in this same professional network, had previously served on the National Council for Health Promotion, contributed to Healthy Israel 2020 and was on the Healthy Cities Advisory Board. Municipal representatives had developed connections with the MOH through Healthy Cities.

At the municipal level, Middle Actors from municipalities, such as health department managers and community center directors collaborated to design nutrition and physical activity programs funded by the NP. Wealthier and/or larger cities, with access to their own funding and significant human resources, had the capacity to devote significant energy to the design and implementation of innovative initiatives. Through forums such as the Healthy Cities Network, the coordinators of these programs were able to present and share lessons learned from these initiatives with municipalities of lower socio-economic status, enabling them to access funding for innovative programs that had been developed elsewhere. On the flipside, municipalities with larger impoverished populations brought extensive experience serving communities with higher levels of chronic disease and obesity, and they brought their expertise to professional forums as well, enabling the sideways sharing of ideas that bred better, more targeted programming.

#### 9.3.2. Mediate

Middle Actors voluntarily launched the NP’s inter-ministerial staff, enabling the program’s ongoing planning, implementation, and evaluation coordination through monthly meetings and routine interactions. They mediated ministerial disputes and planned joint objectives, writing criteria for municipalities, compromising around public relations, and drafting legislation.

When the MOH and the MOE disputed whether to allow processed meat to be served and sold in schools, MOH’s Head Dietician and MOE’s Health Monitor negotiated a compromise, through which no processed food with phosphates would be serve, starting the following year (2017–2018). The negotiation occurred informally, between Middle Actors from each ministry. The resolution entered the final draft of the legislation.

Schools, responsible for implementing the aforementioned WHO module for health promoting schools, had significant leeway for localizing national models for health promotion. Coordinating with the extensive regional staff from the MOE Health Monitor’s Office, each school district, and each individual school enrolled in the initiative, was responsible for designing its own policy changes, programs and parent involvement strategies. The sideways collaboration was dependent upon extensive negotiation between interest groups, such as the regional coordinators, principals, teachers, parents and students, each of which was represented by their own Middle Actors. Ultimately, the successful schools were those where these Middle Actors were able to mediate effectively between interests and seamlessly incorporate health promoting activities such as active recess, healthy snacks, and school gardens into their day-to-day schedules.

#### 9.3.3. Aggregate

All NP-funded partnerships were conditional upon partners adopting Ottawa Charter-styled health promotion and participating in specially designed trainings, spurring additional community partnerships and policy initiatives. The NP’s initiative to implement edible gardens at nursery schools, for example, brought in partnerships with Israel’s leading drip-watering company, which had publicized to middle actors such as leading NGOs and municipalities that it was looking for a corporate responsibility-focused initiative. The company initiated its partnership with edible gardens in one Palestinian-Israeli town in the Coastal North, but subsequently donated equipment to community gardens around the country. In this way, Middle Actors from the NP’s government agencies, municipalities, a leading NGO and industry aggregated funding to incorporate edible gardening in cities and towns statewide.

An additional example: MOCS expanded support for physical activity, through the Sports-for-All Department manager’s position on funding boards and committees, like the Union for School Sports and the National Sports Betting Association. These committees had broadly defined mandates, but board members were given jurisdiction over funding details, creating criteria like raising the number of children participating in physical activity or increasing girls’ representation in sports to funding schemes. In this way, MOCS steered 80 million NIS to women’s sports and added supplemental physical activity in schools for children not in extra-curricular sports programming.

## 10. Discussion: Middle Out, Middle Actors, and Policy Failure

As discussed above, Middle-Out recognizes the importance of intersectoral relationships and Middle Actors’ sensitivity to opportunities, hierarchies and organizational cultures. Enabling, mediating and aggregating are influenced by other actors and context. A Middle-Out lens, therefore, is also useful to clarify when and why actions fail.

While this article mainly focuses on the positive potential of Middle-Out, Middle Actors can also catalyze negative changes or delay positive ones. They can mediate for a less equitable solution or disrupt aggregation.

There were also examples where the limitations of Middle-Out materialized. This included policy failures like limiting television marketing of junk foods to children, and calorie-labeling at restaurants. Documentation of the program between the years 2012 and 2016 reflects Middle-Out-type efforts to overcome obstacles, like leveraging networks with other Middle Actors and supporting policy advancement through social marketing. Still, these policy objectives remained elusive.

A Middle-Out approach was not enough to sustain municipal support amidst budgetary crisis. Middle Actors secured alternative funding sources for smaller municipalities most adversely affected by cuts, but 2 larger municipalities, one in Central Israel and the other in the North, once publicly prominent partners in the NP, severed ties with the program in the aftermath of broken commitments and were openly critical of it.

While incorporating social marketing principals succeeded in bridging research findings and marketing efforts, the attempt to brand the program and develop a uniform marketing language fell short of expectations. While this can be attributed to a funding shortage, higher-level political actors and community representatives actively avoided promoting the brand, even when funding was not an issue, suggesting, perhaps, that Middle Actors’ strengths lie in impacting processes, while others, including ministerial spokespeople, as well as the offices of elected officials and political appointees, may be more effectual when it comes to choosing how processes are presented publicly.

Finally, the “Middle,” is, by definition, contextually defined, a function of positionality in relation to the specific process under analysis. Few actors permanently embody the Top or Bottom, as both can include institutions and individuals [6]. In most cases, a stakeholder at the Top will inevitably find themselves at the Bottom and in the Middle, too, and vice versa. Local leaders, for example, switch from Top Actor, acting and displaying power in a typical Top-Down manner when regulating municipal or neighborhood policies within their own jurisdiction, to Middle Actor, with different capabilities when advocating for their communities in national forums. The Middle, therefore, is a dynamic space.

The power of being in-between, through this lens, is a function of diversity of policy and action networks. At the same time, it is important to stress that not all actors positioned in the ‘Middle’ have the resources, capabilities or the will to act in a Middle-Out manner, or to leverage their Middle-Out potential for the public good. Ultimately, structural and political realities limited Middle Actors, accentuating the need for high-level political will to support many of the changes necessary to foster healthier societies. Effective health promotion clearly involves diverse, interdisciplinary actions by a full mix of actors and sectoral representatives. Middle-Out modes of action are inevitably more powerful when complimented by Top-Down and Bottom-Up. Like Parag and Janda [6] we are not suggesting that Middle-Out replace Top-Down or Bottom-Up. Rather, we argue that the literature is top and bottom-heavy, emphasizing policymaking and individual behavior, privileging actors at more visible extremes. Middle-Out offers a complimentary lens.

## 11. Conclusion: Strengths, Limitations, and Implications for Policy, Practice, and Research

In this article, we illustrated how the Middle-Out perspective draws attention to and helps conceptualize Middle Actors’ contributions to public health “in-between” top-down decision-making and bottom-up grassroots action, including two dimensions of impact: Direction of Influence and Modes of Influence. We utilized Israel’s National Program to Promote Healthy Lifestyle to illustrate the spaces and mechanisms through which these directions of influence can effectively mobilize public health progress, leaning on the actions of the Middle Actors that led the program. We end by noting the study’s limitations and strengths, implications for policy, practice and research, as well as suggestions for further development.

Middle-Out emphasizes professional expertise, longevity and networks. An appreciation for Middle Actor-ship can catalyze increased deference to Middle Actors’ input and recognition of the importance of professional independence. While Israel’s NP had visible Top-Down successes, including legislation and significant budgetary allotments, Middle-Out highlights the contributions that were made possible because of Middle Actors’ access to multiple directions and modes of impact. As a socio-technical perspective, Middle-Out calls attention to the need for public health policy networks that cultivate trust among Middle Actors, as well as the risk of cultivating parallel power structures and engendering competition and territoriality between agencies or departments.

In a political environment in search of results, and given the elusiveness of recognizable public health progress, emphasizing process risks alienating important stakeholders looking for visible “Wins.” Recognizing relevant Middle Actors requires contextualized, case-specific analysis, which policymakers may interpret as an increased bureaucratic burden, a waste of resources. The World Health Organization (WHO) and other institutions that advise health policymaking can increase their emphasis on Middle Actors’ process-sophistication, political savvy, and long-term constancy. They can recommend investing in building durable, intersectoral networks, and mastering the mechanics and political and bureaucratic nuances of implementation.

While this paper focused on effective action from the Middle-Out, just as critical will be analyses that identify a lack of collaborative Middle Actor-ship, where individuals and/or agencies refuse to share information or exclude natural partners from program and policy development. Through this lens, organizational questions such as whether to outsource program management beyond existing networks of Middle Actors grow in importance. As Health in All Policies research grapples with the challenges of implementation, gauging not only when and how collaboration among diverse stakeholders works, but, also, when and how it does not, will be critical. The emphasis on Middle-Out as a successful practice was a methodological limitation which we look forward to remedying with future study.

Research-wise, Middle-Out expands public health inquiry to the nuances of Middle Actor-ship and Middle Actors’ contributions to implementation of Health in All Policies-like initiatives. Epidemiology continues to focus on the reasons for illness or the development of “best-practices” which are often disconnected from their implementation contexts [31]. Increasing the study of professional networks would further elucidate how stakeholders navigate public health’s complexities. Interpretive study and process-sensitive methods like ethnographic fieldwork help achieve this goal. In addition, expanding the study of political will toward investigating the dynamics between supportive politicians, engaged communities and committed Middle Actors would further strengthen collaborative public health practice’s implementation research. Adding Middle Actors to the mix draws attention to how well-meaning politicians, professionals and communities can support one another—a priority that extends to policy arenas well beyond public health.

Empirically, Middle-Out presents challenges. As previously mentioned, one cannot, in most cases, point to a specific actor and categorically define their positionality as The Middle. Indeed, most actors will, at some point, find themselves positioned in-between. Furthermore, the accomplishments of Middle Actors are usually process measures that are harder to identify. Finally, it is simpler to treat institutions as distinct units, as silos that accomplish tasks organizationally. Middle-Out suggests analytically deconstructing departments and organizations and analyzing how, when, and why informal networks transcend organizational boundaries. As Israel’s example shows, Middle-Out redraws allegiances to embody networks that intersect in and out of formal institutions. Each of these attributes—middle Actors’ dynamic relativity, perceived invisibility and institutional flexibility—Can be advantageous in complex socio-technical disciplines such as public health. This ambiguity, though, can complicate empirical analysis. Furthermore, policy networks—even informal ones—can become siloed, privileging specific, familiar partners while marginalizing others.

A final, but meaningful weakness of our study: This research did not discuss whether and how actions from the Middle-Out ultimately impacted health outcomes, and specific health disparities. Nor did it engage the participants’ own socio-economic and demographic makeup. As an ethnographic study, the research focused on a specific group and context, privileging individuals actively involved in the NP’s implementation. It did not focus on those that were excluded, intentionally and unintentionally. Exploring how actors gain access to HiAP’s social spaces would further clarify how society organizes and prioritizes public health and filters the voices that are considered acceptable as possible experts. The possibility that public health’s professional networks empower insiders and marginalize outsiders, points to the need for further inquiry. If Middle-Actor-ship must, analytically, be actively sought out, then researchers’ biases will inevitably impact who gets studied as a Middle Actor and who does not. Continued research can address the pathways through which stakeholders gain the status of Middle Actor, as well as the extent to which the analytical process of inclusion and exclusion produces or reproduces its own set of oppressive hierarchies and practices.

Middle-Out’s limitations accentuate the need for further study in public health contexts and emphasize the importance of continuing to develop complementary approaches for explaining and practicing health governance. As Parag and Janda acknowledge [6], more empirical and critical research is needed to understand Middle-Out and to assess how this approach contributes beyond the existing suite of socio-technical approaches to public health analysis. The urgency that health crises and inequities present should act as the most coherent call for exploring all routes toward equitable population health and incorporating conceptual models developed in fields beyond public health, especially those that elucidate the mechanics of collaborative implementation. Under conditions of political or budgetary instability, or when decision-makers are drawn to other societal challenges or health emergencies, the importance of strong, empowered networks of Middle Actors is heightened. As public health practitioners and scholars encounter the implementation challenges of health governance, Middle Actors’ stability, working relationships and ability to weather storms together while planning for the long term, become indispensable. Understanding their work, therefore, is critical.

## Figures and Tables

**Figure 1 ijerph-16-04993-f001:**
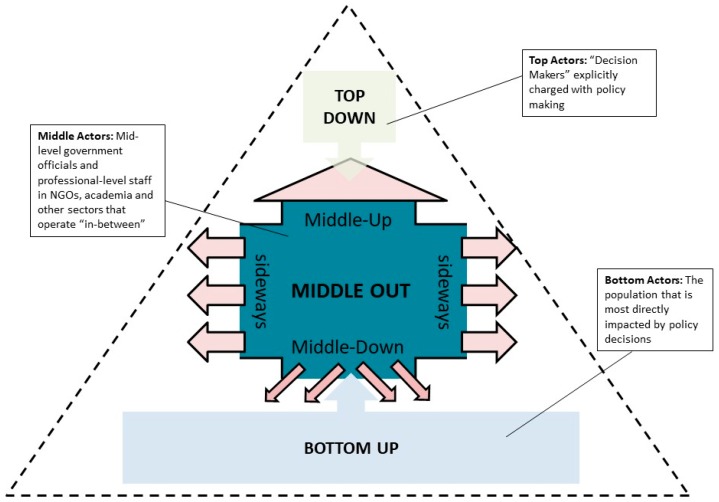
Action from the Middle-Out (adapted from Parag and Janda [6]).

**Table 1 ijerph-16-04993-t001:** Interviewee Information.

Institution	# of Interviewees	Details
Ministry of Health—National Office	8	2 political appointees, 2 senior managers, 3 mid-level managers, 1 entry-level coordinator
Ministry of Health—District Offices	2	2 health promotion fieldworkers
Ministry of Education	1	1 mid-level manager
Ministry of Culture & Sport	3	1 political appointee, 1 senior manager, 1 mid-level manager
Ministry of Agriculture	1	1 mid-level manager
Ministry of Finance	1	1 entry-level analyst
Ministry of Defense	1	1 mid-level manager
Municipalities	2	1 senior manager, 1 mid-level manager
NGOs	3	3 mid-level managers
Academia	3	2 professors, 1 graduate student
Efsharibari Consultants	5	Management, evaluation and communications consultants

**Table 2 ijerph-16-04993-t002:** Middle Out Across the Sectoral Spectrum.

ACTOR	MIDDLE-UP			MIDDLE-DOWN			SIDEWAYS		
	Enable	Mediate	Aggregate	Enable	Mediate	Aggregate	Enable	Mediate	Aggregate
**MID-LEVEL PUBLIC SERVANTS**	Wrote original and subsequent drafts of NP, linking Healthy Israel 2020 recommendations with strategies and partners	Steered program toward low-socio-economic/marginalized communities	Secured funding for partners, spurring their own investments in public health	Diverted existing resources to local NGOs	Negotiated financial and professional support for marginalized communities	Implemented policies like healthier birthdays and banning sugary drinks in nursery schools	Built and sustained intersectoral coalition	Negotiated ministerial contributions and solved disputes	Conditioned partnerships upon partners conducting health promotion trainings for their staffs
**MUNICIPALITIES**	Supported national policymakers with local evidence	Negotiated terms of inter-ministerial collaboration to favor local interests	Formed alliances between cities to influence government amidst budget cuts	Steered national funding to local initiatives and community centers	Localized national policies	Created local coalitions, like healthy workplace forums	Wealthier municipalities developed programs later utilized by poorer ones	Chose most effective representatives for leading the program’s local implementation	Required municipal departments to integrate health promotion in strategic planning
**SCHOOLS**	Lobbied for government certification for health-promoting schools	Translated national standards to school needs	Coordinated with food suppliers and parents’ unions to shape nutrition legislation	Opened sports facilities to public after school hours	Involved parents/community	Coordinated healthy birthdays among all families	Nursery schools participated with municipalities in healthier foods pilot	Designed curricula for integrating health content into math, science and other subjects	Expanded programs that worked in one class/grade to others
**ACADEMIA**	Contributed research for decision-making, like Health Behavior in School Children study	Critiqued status quo in Parliament/media	Engaged additional academic partners in advocacy	Helped grassroots initiatives obtain funding	Social marketing team translated epidemiology into tangible materials	Facilitated unified voice for groups like parents of young children	Built bridges across sectoral and geographic divides	Participated in program’s professional forums	Incentivized students to work on program, including graphic design, surveys and marketing

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
