# Peer review of "Public Health from the Middle-Out: A New Analytical Perspective"

_ijerph, 2019, doi:10.3390/ijerph16244993_

Round 1

Reviewer 1 Report

Comments have been addressed

Author Response

Thanks very much for your request that we proof read the manuscript. We have done so, and believe we have corrected any outstanding errors.

Reviewer 2 Report

I still am missing the explanation whether the observations were done with a guide about the items/issues to observe. I did not find that information in the manuscript. Perhaps 1 sentence can be added to explain this to the paragraph on lines 5-10, page 6.

Otherwise, very innovative and interesting approach and perspective for the study. I find this to be of great interest to the readers.

Author Response

Thanks very much for your support of our manuscript! At your suggestion, we have added several sentences of explanation on page 6 (marked with a "comment"), which describe fieldnote-taking practices, as well as thematic emphases.

This manuscript is a resubmission of an earlier submission. The following is a list of the peer review reports and author responses from that submission.

Round 1

Reviewer 1 Report

There are a number of worrying aspects to this paper that did not help my understanding of the ‘how’ in PH. Consistency and relevance to PH are core aspects that should be reviewed by the authors if they decide to resubmit.

Author Response

Thank you for this critique. In this revised draft, we have attempted to increase our linguistic and conceptual consistency, focus the background/introduction, be more articulate with our methodology and emphasize the relevance of the paper to public health research and practice. While we have tried to address these points throughout the draft, we hope that you will find that the revised abstract, results, discussion and conclusion sections clarify the paper’s public health significance and address your “how”-related questions.

Reviewer 2 Report

August, 2019/ Peer review – manuscript ijerph-564413

I found this article manuscript very interesting. It is a useful contribution in highlighting the role of the mid-level stakeholders in the policy making process and their influence through the multi-level governance perspective described well in the introduction part of the manuscript.

I suggest publishing after some methodological weaknesses are addressed properly:

There is lacking information about what was observed in the selected multiple settings (p. 5, rows 197-202) particularly, by who and how. Was there a kind of protocol that was used to make the observations or were the observations just ad hoc. How was the data coming from the observations dealt with or analysed and how this data was integrated in the entire data analysis? The sampling for the interviews is well described (p. 5, rows 203-211), but the conduct of interviews were not described at all. The number of interviewees was mentioned to be 20, but there is no information whether these people were policy makers, professionals in the filed or lay people, etc. It would be good to have a description of the informants, how many men and women. Were the interviews conducted with an interview guide and how were the themes of the guide composed. The documents that were included were listed, but there is a lack of information how the documents were analysed. Were they included in the Narralizer, e.g., and analysed in the similar manner as the interviews? How the different types of data were integrated in the analysis is not described in the methods chapter. It would be necessary to describe in more detail what the Directive Thematic Analysis included. This method has generally been described as a deductive method, that supports some existing theory. Thus, it would be good the write out what were the starting categories or initial codes that were used in the beginning of the analysis and did the further analysis reveal new emerging codes? What were the distinguished steps of the analysis process? Perhaps that process could be illustrated by a figure. What processes were used to ensure the trustworthiness of the results and their neutrality or unbiassness?

Author Response

Point 1: There is lacking information about what was observed in the selected multiple settings (p. 5, rows 197-202) particularly, by who and how. Was there a kind of protocol that was used to make the observations or were the observations just ad hoc. How was the data coming from the observations dealt with or analysed and how this data was integrated in the entire data analysis?

Response 1: Thank you for requesting this information. We believe deeply in the importance of participatory observation to analysis of public health practice and were, therefore, excited to elaborate further. We have included information in the methods section (page 5) that addresses, we hope, the gaps that you outlined around both the collection and analysis of observational data.

Point 2: The sampling for the interviews is well described (p. 5, rows 203-211), but the conduct of interviews were not described at all. The number of interviewees was mentioned to be 20, but there is no information whether these people were policy makers, professionals in the filed or lay people, etc. It would be good to have a description of the informants, how many men and women. Were the interviews conducted with an interview guide and how were the themes of the guide composed.

Response 2: We agree that this information should be included. We have added a table (labelled table 1 on page 5) which includes additional information about the interviewees, such as institutional affiliation and hierarchical position, included the gender breakdown, as well details regarding interview guide and the emergence of themes.

Point 3: The documents that were included were listed, but there is a lack of information how the documents were analysed. Were they included in the Narralizer, e.g., and analysed in the similar manner as the interviews?

Response 3: We have added this information (page 6). The data utilized for this research, including fieldnotes, interview transcripts and documents, were included in the Narralizer and analysed in a similar manner to the interviews.  

Point 4: How the different types of data were integrated in the analysis is not described in the methods chapter. It would be necessary to describe in more detail what the Directive Thematic Analysis included. This method has generally been described as a deductive method, that supports some existing theory. Thus, it would be good the write out what were the starting categories or initial codes that were used in the beginning of the analysis and did the further analysis reveal new emerging codes? What were the distinguished steps of the analysis process? Perhaps that process could be illustrated by a figure.

Response 4: Directive Thematic Analysis, we believe, was an appropriate analytical technique for this work, as it enables establishing conceptual boundaries that serve the aims of the research, in this case, informing public health research and practice, as well as leaving room for interpretation and emergent themes. We have added information on to the methods section (see page 6) regarding our own analytical process, including the formation of a priori codes and the transition to emergent codes, providing examples of each code type. We agree with you that this information should be included and appreciate your guiding questions, which directed our revisions.

Point 5: What processes were used to ensure the trustworthiness of the results and their neutrality or unbiassness?

Response 5: To ensure trustworthiness, we combined peer debriefing with intertextuality and triangulation. We added a description of these processes to the methods section, the first paragraph on page 7. Thank you for requesting this information.

Reviewer 3 Report

Thank you for a well written article regarding the application of a novel approach to understanding the role of 'middle actors' as agents of change within public health.

In its current format, the background section of the article provides an overly extensive description of the Middle-Out approach. It is suggested that this section of the article be rewritten and shortened to provide an appropriate overview rather than what appears to be a paraphrasing of 2 key references.

The study design and methods are well described. The results and discussion sections are well structured and presented in table 1. This could benefit from being titled as results with relevant sub-headings. Again there appears to be additional description of the Middle-Out approach provided in an ad hoc manner rather than as part of the discussion of the results. 

Author Response

Point 1: In its current format, the background section of the article provides an overly extensive description of the Middle-Out approach. It is suggested that this section of the article be rewritten and shortened to provide an appropriate overview rather than what appears to be a paraphrasing of 2 key references.

Response 1: Agreed – we should have been more concise here. Extensively paraphrasing two key references was not our intention; we’re grateful to you for pointing out this issue. We have revised this background section, cutting its length in half and incorporating relevant content in the results, discussion and conclusion sections.

Point 2: The results and discussion sections are well structured and presented in table 1. This could benefit from being titled as results with relevant sub-headings. Again there appears to be additional description of the Middle-Out approach provided in an ad hoc manner rather than as part of the discussion of the results.

Response 2: Thank you for this comment. We have re-titled these sections (Results, Discussion, Conclusion). We have left the results section sub-heads organized according to the categories in the table, including the Directions of Influence (Upstream, Downstream and Sideways) and the Modes of Influence (Enable, Mediate and Aggregate), in order to provide consistency with the background section. 

Round 2

Reviewer 1 Report

This paper still has some worrying aspects as follows. 

Bottom-up and top-down are recognised terms that are used in PH and are clearly separate from an up and down stream discourse. It is important for the authors to situate their discussion within the present dialogue and to include the origins of the key related (top down/bottom up) terms. The paper runs the risk that by introducing a new perspective that has been adapted to suit the purposes of the discussion in the paper the authors will confuse rather than clarify the issue.

There are a number of areas of contradiction in the terms, for example, ‘organized citizen groups, professional associations, faith-based organizations’ are all part of civil society and at the ‘bottom’ as they represent the interests of people. And, middle actors do not necessarily ‘have interactions both at the population level, as well as with higher level decision-makers’ as they are often civil servants and bureaucrats that are employed to deliver services, information and resources and by the same system as those at the ‘top’.

Figure 1 is helpful to the reader but table 1 is not as it discusses up and down stream and deviates away from the key terms used. The paper reads as an endorsement of the model presented by Parag and Janda rather than a critique of its relevance to PH. This is another worrying aspect and I did not find the paper helped my understanding of the ‘how’ in PH. Consistency and relevance to PH are core aspects that should be reviewed by the authors if they decide to resubmit.

Reviewer 3 Report

A much improved manuscript.

It may also be useful to include strengths and limitations of your study within the discussion.